# What Makes a Representation Relightable? Probing Visual Priors via Augmented Latent Intrinsics

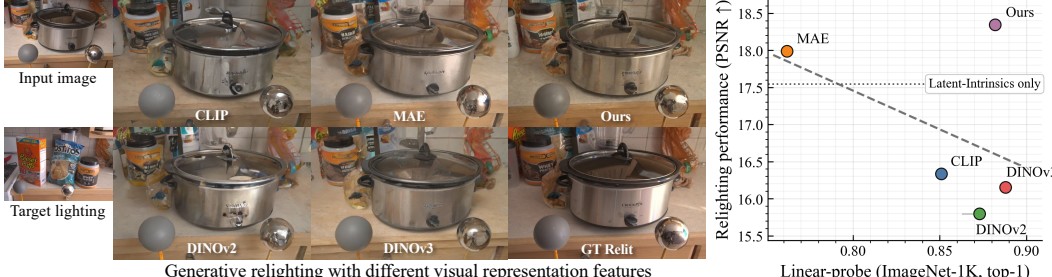

Figure 1: **Stronger Semantic Encoders Can Harm Relighting Performance. Left:** Visual comparison on a scene with complex specular materials. The task is to relight the input image (top-left) using the target illumination (bottom-left), which requires moving specular highlights from left to right, as indicated by the chrome sphere. While features from semantic encoders (CLIP, DINO) fail to reproduce realistic highlights, the MAE plausibly moves the highlight but blurs fine details, such as text labels. Our method (top-right), which combines features from RADIO (a pretrained model; distilled from many vision encoders) with latent intrinsics, closely matches the ground truth. **Right:** Quantitative analysis reveals a trade-off: for most encoders optimized for pure semantics, relighting quality (PSNR) is inversely correlated with recognition performance (ImageNet-1K linear probing as reported in the original papers.). Our approach breaks this trend, achieving high performance on both tasks.

## Abstract

Image-to-image relighting requires a representation that disentangles scene properties from illumination. Recent methods use latent intrinsic representations but remain under-constrained and often fail on challenging materials like metal and glass. A natural hypothesis is that injecting powerful, pretrained visual priors should resolve these failures. We find the opposite is true: features from top-performing semantic encoders often degrade relighting quality, revealing a fundamental trade-off between semantic abstraction and photometric fidelity. This paper investigates what makes a representation "relightable." We introduce Augmented Latent Intrinsics (ALI), a method that resolves this trade-off by strategically fusing features from a dense, pixel-aligned visual encoder into a latent-intrinsic framework while leveraging self-supervision refinement to overcome the scarcity of paired real-world training data. Trained only on unlabeled, real-world image pairs, ALI achieves strong relighting improvements with the largest gains on complex materials.

## 1 Introduction

Image-based relighting is the task of transferring illumination from one image to another. It has important applications in augmented reality, computational photography, and digital content creation. Yet it remains ill-posed: brightness can come from a light source or a shiny surface, and darkness can signal shadow or absorption. Any successful model must reason about both scene semantics and how light interacts with scene geometry and material properties.

Traditionally, relighting is treated as a graphics problem. In this paper, we take a different view: relighting can also be a probe to understand what visual representations are encoded about the

physical world. Unlike recognition tasks, which reward semantic abstraction, relighting demands a representation that can simultaneously disentangle illumination from scene properties while preserving the fine-grained material and geometric cues necessary to render new lighting effects. This creates a trade-off between semantic abstraction and physical fidelity, offering a novel perspective on representation learning beyond classification. And our goal is to improve relighting by integrating visual priors.

We specifically focus on image-to-image relighting. Unlike classical inverse-graphics pipelines, which explicitly estimate geometry, reflectance, and light sources, we ask whether learned features, when augmented with latent intrinsic representations (Zhang et al., 2024a), are sufficient to support illumination transfer. This direct setting avoids explicit inverse-graphics decompositions but puts strong demands on the representation. While recent diffusion-based models show promise, they often fail on materials with complex, view-dependent effects like specularity and transparency (Xing et al., 2025; Zeng et al., 2024b; Liang et al., 2025; He et al., 2025).

A natural hypothesis is that large-scale pretrained visual representations (e.g., from CLIP, DINO), should resolve these ambiguities. Our findings reveal a more complex reality. Features from encoders like CLIP and DINO, which are optimized for semantic invariance via contrastive or distillation losses, systematically degrade relighting performance. Conversely, features from a masked auto-encoder (MAE)—a model trained on a dense pixel reconstruction task—yield better results despite a weaker semantic score (Fig. 1). This reveals a critical gap: the very inductive biases that create powerful semantic representations appear to discard the fine-grained photometric and spatial information required for physical reasoning. The relative success of a reconstructive model like MAE suggests that preserving this dense, pixel-level structure is crucial.

This raises our core research question: What makes a representation *relightable*? We define this as the ability to achieve (i) photometric consistency under light transfer, (ii) material fidelity for view- and material-dependent effects, and (iii) illumination invariance while retaining per-pixel detail. Our analysis reveals a clear trade-off. High-level semantic features discard the dense, pixel-aligned information required for accurate light transport. Conversely, low-level features lack the semantic context to handle material boundaries correctly. We find the most effective foundation comes from dense, pixel-aligned encoders like RADIO (Heinrich et al., 2024b), which retain spatial structure while carrying semantic signals. This is because RADIO is distilled from multiple teachers (DINO, SigLIP, MAE, etc.), to carry semantic signals and also preserve dense cues and long-range light-surface interactions.

Based on this insight, we introduce Augmented Latent Intrinsics (ALI), a method designed to balance this semantic–photometric trade-off. ALI consists of: (i) a fusion adapter that couples latent intrinsic features with semantic features from a visual encoder, (ii) a progressive training schedule that aligns the generative decoder with the newly introduced semantic features in ALI, and (iii) a self-supervision loop to circumvent the lack of a large-scale real-world multi-illumination dataset. Trained only on unlabeled real-world image pairs – a key advantage over prior work requiring large synthetic datasets (Zeng et al., 2024b; Liang et al., 2025; He et al., 2025) – our ALI model establishes a new state-of-the-art on the MIIW benchmark, demonstrating significant improvements on glossy and specular materials. These findings suggest that progress in generative relighting can be made not from scaling alone, but from *systematic evaluation* and *targeted fusion* of complementary priors.

In summary, our contributions are

- A controlled protocol and operational criteria for probing *relightability*, explaining why popular semantic encoders fail despite strong recognition performance.

- **ALI**: a lightweight fusion adapter with progressive training and self-supervision that fuses a visual encoder with latent intrinsics to balance semantics and dense photometry.

- A refinement protocol that overcomes the scarcity of paired real-world training data and does not require any synthetic or computer graphics labels for training.

- State-of-the-art results on MIIW among open-sourced diffusion-based methods, improving RMSE by **4.5%** and SSIM by **4.9%** over *LumiNet*, with the largest gains on glossy and specular materials.

## 2 RELATED WORK

**Generative Relighting.** We focus on image-to-image relighting, which requires disentangling illumination from intrinsic scene properties from a single monocular image. While traditional approaches relied on inverse rendering with known geometry or lighting (Li et al., 2022; 2023; 2021; Zhang et al., 2016; Zhu et al., 2023; Garon et al., 2019; Gardner et al., 2019), recent generative models have achieved impressive results on objects (Zeng et al., 2024a; Deng et al., 2024; Jin et al., 2024; Zhang et al., 2025a; Bharadwaj et al., 2024), portraits (Pandey et al., 2021; He et al., 2024; Mei et al., 2025), and indoor scenes (Kocsis et al., 2024a; Bhattad et al., 2024; Xing et al., 2025; 2024; Choi et al., 2025; Zeng et al., 2024b). However, many of these methods, including concurrent works like *LightLab* (Magar et al., 2025), *UniRelight* (He et al., 2025), *IntrinsicEdit* (Lyu et al., 2025), depend on dense supervision or synthetic data, limiting their real-world applicability.

An alternative line of work achieves unsupervised relighting by leveraging priors from pretrained generative models (Phongthawee et al., 2024). Methods like *StyLitGAN* (Bhattad et al., 2024) and *LumiNet* (Xing et al., 2025) use inductive biases in GANs and diffusion models, but their reliance on purely photometric losses leaves them vulnerable to intrinsic ambiguities, especially for complex materials where semantic guidance is needed.

**Intrinsic Images.** Intrinsic image decomposition aims to disentangle an image into components like albedo and shading (Barrow & Tenenbaum, 1978). The field has a long history, with methods using low-level cues (Baslamisli et al., 2021; Luo et al., 2020; Das et al., 2022; Fan et al., 2018; Chen & Koltun, 2013; Xing et al., 2022), physical assumptions (Barron & Malik, 2014; Grosse et al., 2009), ordinal supervision (Careaga & Aksoy, 2023; 2024; Dille et al., 2024), or semantics (Baslamisli et al., 2018). More recently, generative priors have been used for in-the-wild generalization (Bhattad et al., 2023; Du et al., 2023; Kocsis et al., 2024b; Zeng et al., 2024b; Luo et al., 2024; Xi et al., 2024), though often still relying on synthetic data. Self-supervised approaches instead use real image pairs under different lighting (Li & Snavely, 2018; Ma et al., 2018; Janner et al., 2017). *Latent Intrinsics* (Zhang et al., 2024a) extends this to latent space without labels, but the resulting representations lack the high-level semantic grounding needed to resolve material and illumination ambiguities in relighting.

**Visual Representations.** Self-supervised learning on unlabeled images has produced powerful visual encoders through either discriminative objectives that align augmented views (He et al., 2020; Chen et al., 2020; Caron et al., 2021) or generative objectives that reconstruct corrupted inputs (Vincent et al., 2008; Ho et al., 2020; He et al., 2022a; Zhang et al., 2024b;c). However, most are optimized for high-level recognition tasks, whereas relighting requires features that span both low-level photometric cues and high-level structure. To our knowledge, this is the first work to apply large-scale visual representations to generative image relighting. We show that fusing features from a distilled, pixel-aligned encoder like RADIOv2.5H (Heinrich et al., 2024b; Fang et al., 2023; Radford et al., 2021; Zhai et al., 2023; Oquab et al., 2023; Kirillov et al., 2023; Xiao et al., 2024) with latent intrinsic representations yields semantically grounded, physically consistent light transfer.

## 3 PRELIMINARIES

Our work builds on the concept of latent intrinsics, where lighting-invariant features can be learned from multi-illumination image pairs without direct supervision (Zhang et al., 2024a). Given an image $I_s^l$ of a scene $s$ under lighting $l$, an encoder $E_\theta$ disentangles it into a set of hierarchical, lighting-invariant intrinsic features $\{S_{s,i}^l\}$ and a global lighting embedding $L_s^l$. A decoder $D_\phi$ can then relight the scene by combining the intrinsics from one view with the lighting from another:

$$\widetilde{I}_s^{l_1 \to l_2} = D_\phi(\{S_{s,i}^{l_1}\}, L_s^{l_2}). \tag{1}$$

The model is trained from scratch using a combination of reconstruction, intrinsic invariance, and latent space regularization losses.

However, because these models are typically trained on a limited set of real-world multi-illumination pairs, their learned representations are constrained to the lighting and material types seen during training. This often leads to failures in accurately disentangling complex materials under diverse, unseen lighting conditions. To address this, we propose to augment the latent intrinsic features with powerful visual priors from foundation models trained on large-scale, diverse image collections.

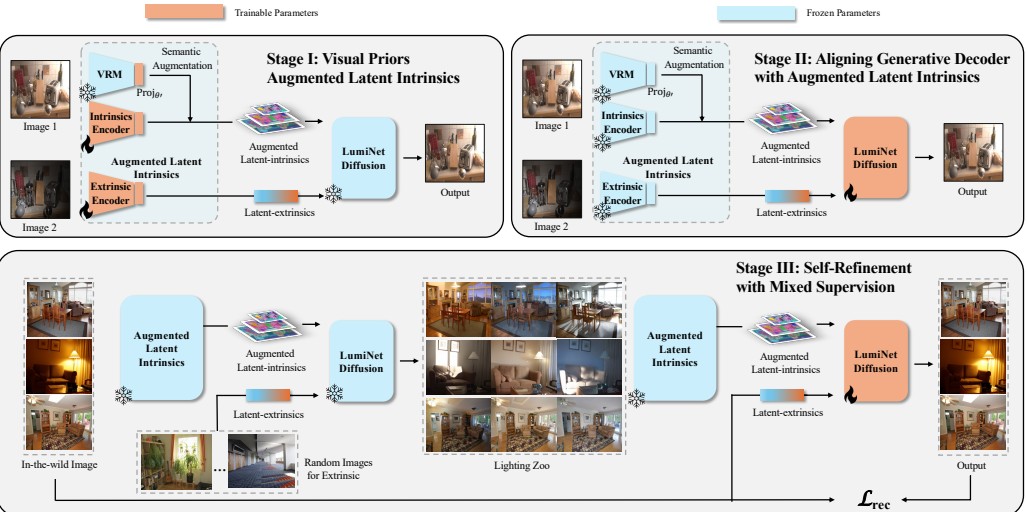

Figure 2: **Our Three-Stage Training Pipeline for Augmented Latent Intrinsics (ALI).** Our method progressively adapts a pretrained visual encoder and fine-tunes a generative decoder for high-fidelity, unsupervised relighting. **Stage I: Augmenting Latent Intrinsics.** We inject semantic features from a frozen vision encoder into the intrinsics encoder. This creates our semantically-enriched **ALI**, which better disentangles scene properties from illumination. **Stage II: Aligning the Generative Decoder.** With the encoder fixed, we fine-tune the LumiNet diffusion decoder to condition on the new ALI representation, aligning the generator with the scene's improved physical understanding. **Stage III: Self-Refinement.** We generate a "Lighting Zoo" (examples in supplementary) of pseudo-relit images to overcome data scarcity. These synthetic images serve as new inputs, with the original real image as the ground truth. This self-supervision trains the network to ignore artifacts and focus on essential structural properties, significantly improving realism and robustness on in-the-wild images.

While these priors are effective for high-level tasks, their utility for the fine-grained physical reasoning required in relighting remains unexplored.

## 4 METHOD

Our goal is to leverage powerful, pretrained visual priors for image-to-image relighting. We adapt these priors for use in a latent-intrinsic model, specifically LumiNet (Xing et al., 2025), through a three-stage training strategy.

**Stage 1: Augmenting Latent Intrinsics.** To improve the disentanglement of latent intrinsic (like albedo) and extrinsic (like lighting), we inject semantic context from a frozen visual encoder $\boldsymbol{E}_{\text{sem}}$ (e.g., RADIOv2.5) into the latent intrinsics. First, we extract a hierarchy of feature maps $\{\boldsymbol{V}_{s,i}^l\}_{i=1}^N = \boldsymbol{E}_{\text{sem}}(\boldsymbol{I}_s^l)$ from an input image $\boldsymbol{I}_s^l$, where $N$ denotes the number of selected intermediate layers from each VRM encoder; the exact layer indices and feature dimensions are provided in the supplementary material. These maps are upsampled to the input resolution and concatenated into a pixel-wise hypercolumn descriptor $\boldsymbol{H}_s^l(x, y)$. A learnable projection layer $\text{Proj}_{\theta'}$ then aligns these features with the intrinsic features $\boldsymbol{S}_{s,i}^l$ from the relighting encoder $\boldsymbol{E}_\theta$:

$$\boldsymbol{A}_{s,i}^l = \text{Proj}_{\theta'}(\boldsymbol{H}_{s,i}^l) + \boldsymbol{S}_{s,i}^l. \tag{2}$$

In this stage, we freeze the visual encoder $\boldsymbol{E}_{\text{sem}}$ and all decoders, training only the relighting encoder $\boldsymbol{E}_\theta$ and the projection layers $\text{Proj}_{\theta'}$. The training objective combines a standard reconstruction loss $\mathcal{L}_{\text{relight}}$ and a hyperspherical regularization loss $\mathcal{L}_{\text{reg}}$ applied to both intrinsic and lighting features (Zhang et al., 2024a):

$$\mathcal{L}_{\text{reg}}(\boldsymbol{A}) = \left\| R(\boldsymbol{A}) - R(\widehat{\boldsymbol{A}}) \right\|_2^2 \tag{3}$$

$$R(\boldsymbol{A}) = \log \det \left( \boldsymbol{I} + \frac{d}{n\lambda^2} \boldsymbol{A}^\top \boldsymbol{A} \right), \tag{4}$$

where $\widehat{A}$ is sampled from a uniform hyperspherical distribution, $n$ is the number of spatial locations, $d$ is the feature dimension, and $\lambda$ is a regularization temperature parameter. This term encourages features to uniformly spread out in the sphere, aiding feature optimization.

To stabilize training, we replace the original intrinsic invariance loss with an improved version that regularizes the intrinsic features $\boldsymbol{A}_{s,i}^{l_n}$ of a scene $s$ toward their mean across $M$ different lighting conditions:

$$\mathcal{L}_{\text{Improved Intrinsics}} = \sum_{s,m} \| \boldsymbol{A}_{s,i}^{l_m} - \frac{1}{M} \sum_{m'} \boldsymbol{A}_{s,i}^{l_{m'}} \|_2. \tag{5}$$

The total loss for this stage is $\mathcal{L}_{\text{Stage1}} = \mathcal{L}_{\text{relight}} + \mathcal{L}_{\text{Improved Intrinsics}} + \sum_i \mathcal{L}_{\text{reg}}(\boldsymbol{A}_{s,i}^l) + \mathcal{L}_{\text{reg}}(\boldsymbol{L}_s^l)$.

**Stage 2: Aligning the Generative Decoder.** With visual encoder $E_{\text{sem}}$ relighting encoder $E_\theta$ and projection layer $\text{Proj}_{\theta'}$ fixed, we fine-tune LumiNet's diffusion decoder $\boldsymbol{D}_{\phi'}$ to align with our semantically-augmented latent intrinsic representation $\boldsymbol{A}_{s,i}^{l_1}$. The decoder is trained to predict the noise $\epsilon$ added to the target lighting latent $\boldsymbol{L}_s^{l_2}$ at timestep $t$, conditioned on the new intrinsics. The optimization uses only the standard denoising score-matching loss from DDPM (Ho et al., 2020):

$$\mathcal{L}_{\text{Stage2}} = \mathbb{E}_{t,\epsilon} \left[ \| \boldsymbol{D}_{\phi'}(\alpha_t \boldsymbol{L}_s^{l_2} + \beta_t \epsilon, \{\boldsymbol{A}_{s,i}^{l_1}\}, \boldsymbol{L}_s^{l_2}) - \epsilon \|_2^2 \right] \tag{6}$$

**Stage 3: Self-Refinement.** To overcome the scarcity of paired real-world data, we further refine the decoder using an iterative self-training scheme. We generate pseudo-relit image pairs by transferring illumination between random images within a batch. The decoder is then fine-tuned on these pseudo-pairs using the same diffusion loss as in Stage 2. This process is regularized by mixing in occasional real-world same-image reconstructions (where source equals target) to ensure content fidelity. This self-refinement progressively improves realism and robustness without requiring any labeled data.

## 5 EXPERIMENTS

**Training Datasets.** For the first two stages, we train **ALI** on real-world image pairs from two datasets: the *MIT Multi-Illuminant* (MIIW) dataset (Murmann et al., 2019), containing 985 scenes under 25 lighting conditions, and the *BigTime* dataset (Li & Snavely, 2018), with 460 scenes under 20–50 natural illuminations.

**Training Procedure.** Our three-stage process is optimized using AdamW Loshchilov & Hutter (2019) with a learning rate of $4 \times 10^{-5}$. In **Stage I**, we use a scene-aware batch sampler that ensures each batch contains multiple views of the same scene, enforcing consistency for the learned intrinsic representations. For **Stage II**, we relax this constraint and sample images across different scenes to promote diversity in the generative decoder. Finally, **Stage III** focuses on self-refinement to improve generalization. We randomly sample 1,000 scenes from our lighting zoo to generate pseudo-relit data. Please refer to Sec. A.2 for details on the generation of the lighting zoo.

Interestingly, while this stage does not yield further improvements on the MIIW test set, it consistently enhances performance on in-the-wild images, particularly in preserving structural fidelity.

**Inference.** Even though our model is trained on image pairs from the same scene, at inference, it generalizes to relighting arbitrary unpaired images. The intrinsic features are extracted from a content image, while the lighting embedding is taken from a separate illumination image and transferred via the diffusion decoder. Following LumiNet, we adopt the bypass decoder (Wang et al.) by default during inference to better preserve the identity of the generated images. However, all feature fusion experiments are conducted without the bypass decoder to ensure that the performance of the relit images solely reflects the quality of the augmented latent intrinsic representation.

**Evaluation.** Our evaluation is two-fold. First, we demonstrate the effectiveness of the proposed representation through its superior generative relighting performance. Then, we analyze why the representation is beneficial by evaluating relighting consistency across different material categories. Additionally, we conduct a thorough ablation study for each experimental configuration.

**Cross-Scene Relighting.** We evaluate our method on the unseen test split of the MIIW dataset and compare it with both feed-forward approaches, including SA-AE (Hu et al., 2020) and Latent-Intrinsic (Zhang et al., 2024a), as well as diffusion-based models such as RGB↔X (Zeng et al., 2024b)

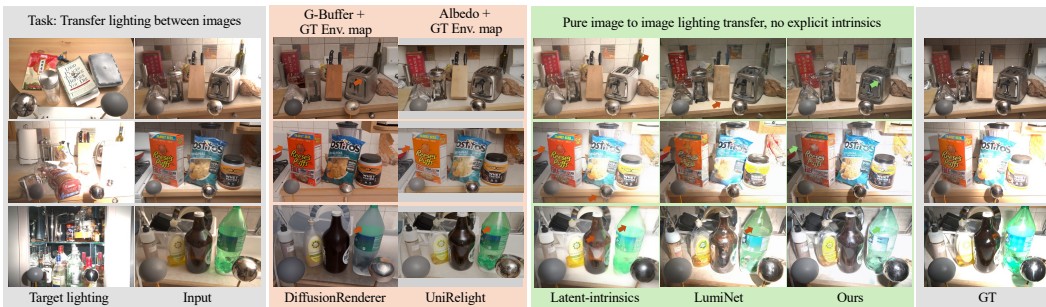

Figure 3: **Qualitative comparison of relighting methods on challenging MIIW test scenes.** Our image-to-image approach produces more physically plausible results than competing methods, many of which rely on privileged information like GT light maps, G-buffers, or albedo. The task is to relight the *Input* scene using the illumination from the *Target lighting* image. In the **top row**, competing methods render the metallic toaster with faint or blurry specular highlights, while our method produces sharp, plausible reflections that are much closer to the ground truth (GT). Similarly, in the **second row**, all baseline methods generate incorrect or missing shadows of the orange cereal box, and `LumiNet` also blurs the text on the packaging. Our method preserves these details and renders more realistic shadows. Finally, in the **third row**, all baselines struggle to render the complex transparency and caustic light effects of the bottles, a common failure case that our method addresses with a more plausible result. The results for `UniRelight` were provided by the authors. Our approach demonstrates that augmenting dense visual priors with latent intrinsics improves a model's relightability and its understanding of light transport, all without 3D or inverse graphics supervision. Red and green markers highlight specific failures in competing methods and successes in our method, respectively. *Best viewed on screen with zoom.*

and LumiNet (Xing et al., 2025). Following the standard protocol, each trial randomly samples one target image and 12 reference lighting conditions (Zhang et al., 2024a). We report performance on both the raw model outputs and the results after global color correction, which compensates for white-balance discrepancies (Zhang et al., 2024a). As shown in Tab. 1, our method achieves state-of-the-art performance among diffusion-based approaches under both evaluation settings, surpassing existing diffusion models by a substantial margin—yielding +4.9% SSIM and +4.5% RMSE improvement. Note that our method does not outperform models trained exclusively on MIIW, such as SA-AE (Hu et al., 2020) and Latent-Intrinsic (Zhang et al., 2024a), because conventional quantitative metrics primarily emphasize tone consistency rather than accurate modeling of shadows or specularities, a trend that is clearly reflected in the qualitative results (Fig. 3) and human evaluation (Tab. 3).

For qualitative comparisons (Fig. 3), we benchmark against state-of-the-art models, noting that several require privileged information: `DiffusionRenderer` was provided with the GT light probe from the MIIW dataset, and the results for `UniRelight` (sourced from the authors) require albedo and the light probe / environment map. As shown, `LumiNet` achieves coherent lighting but struggles to preserve material and geometric details, such as on the toaster (row 1) and bottles (row 3), highlighting the limitations of traditional latent-intrinsic frameworks. In Fig. S.2, we demonstrate same-scene relighting, where our method consistently produces materially consistent results on ambiguous surfaces like glass and metal, while `RGB↔X` and `LumiNet` often yield distortions.

**In-Scene Relighting.** We also evaluate *in-scene* relighting on MIIW (Tab. 2), where for each scene we select an image captured under illumination $i$ and relight it to the corresponding $(i + 12)$ lighting condition. As reported in Tab. 2, our method exhibits consistent performance and achieves the second-best LPIPS among all methods, without requiring any additional privileged labels. A concurrent work, UniRelight, despite being trained with privileged labels, leveraging a video model for relighting, and being specifically trained on the MIIW dataset, still struggles to produce accurate relighting, as further evidenced by the qualitative comparisons (Fig. 3).

**Evaluation on In-the-Wild Images.** In Fig. 4, we present a qualitative comparison against IC-Light (Zhang et al., 2025a), DiffusionRenderer (Liang et al., 2025), Latent Intrinsics Zhang et al. (2024a), and LumiNet (Xing et al., 2025) . Our model produces relit images that more faithfully match the target illumination, capturing subtle shadows and consistent reflectance. In contrast, IC-Light

often suffers from oversaturated and artisitic highlights, and LumiNet introduces noticeable artifacts on reflective surfaces and also fail to get fine details and thin structures.

Table 1: **MIIW *cross-scene* evaluation.** Following Latent-Intrinsics and LumiNet, this protocol compares source and target images from different scenes. We report RMSE and SSIM on raw and color-corrected outputs. Trained on diverse images without privileged labels, our method performs competitively—especially within its training category. **Bold** highlights the best in each block (MIIW-only vs. diverse images for training). Note that metrics favor tone consistency over shadow or specular accuracy; qualitative results remain the primary evidence of lighting fidelity (Giroux et al., 2024).

| Methods | Labels | RMSE↓ | SSIM↑ | RMSE↓ | SSIM↑ |
|---|---|---|---|---|---|
| **Trained only on MIIW** | | | | | |
| SA–AE (Hu et al., 2020) | Light | 0.288 | 0.484 | 0.232 | 0.559 |
| SA–AE Hu et al. (2020) | – | 0.443 | 0.300 | 0.317 | 0.431 |
| S3Net (Yang et al., 2021) | Depth | 0.512 | 0.331 | 0.418 | 0.374 |
| S3Net (Yang et al., 2021) | – | 0.499 | 0.336 | 0.414 | 0.377 |
| Latent–Intrinsic (Zhang et al., 2024a) | – | 0.297 | 0.473 | **0.222** | **0.571** |
| **Trained on diverse indoor images** | | | | | |
| RGB↔X (Zeng et al., 2024b) | G-Buffer | 0.587 | 0.070 | 0.427 | 0.215 |
| DiffusionRenderer (Liang et al., 2025) | Env. map | 0.399 | 0.354 | 0.341 | 0.355 |
| LumiNet (Xing et al., 2025) | – | 0.310 | 0.440 | 0.240 | 0.527 |
| **Ours** | – | **0.294** | **0.464** | **0.231** | **0.553** |

Table 2: **MIIW *in-scene* evaluation.** Results on the multi-illumination dataset of Murmann et al. (2019), where source and target are the same scene captured under different illuminations. We report PSNR, RMSE, LPIPS, and SSIM. Our method, trained on diverse images without privileged labels, achieves competitive results compared to prior approaches. Higher PSNR/SSIM and lower RMSE/LPIPS indicate better performance. [†] trained only on MIIW; [*] numbers reported from UniRelight. Pixel metrics can be sensitive to small misalignments (Giroux et al., 2024).

| Methods | Labels | PSNR↑ | RMSE↓ | LPIPS↓ | SSIM↑ |
|---|---|---|---|---|---|
| RGB↔X (Zeng et al., 2024b) | G-Buffer | 15.674 | 0.161 | 0.323 | 0.500 |
| DiffusionRenderer (Liang et al., 2025) | Env. map | 16.810 | 0.156 | 0.343 | 0.612 |
| UniRelight (He et al., 2025)[*] | Albedo, Env. map | 20.760 | – | 0.251 | 0.749 |
| Latent-Intrinsics (Zhang et al., 2024a)[†] | – | **21.350** | **0.092** | **0.157** | **0.794** |
| LumiNet (Xing et al., 2025) | – | 18.568 | 0.123 | 0.228 | 0.645 |
| **Ours** | – | 18.872 | 0.119 | 0.213 | 0.671 |

Table 3: **User Study**. We conduct two user studies: (1) a comparison of in-the-wild performance among LumiNet, Latent-Intrinsics, and our method; and (2) a stage-wise evaluation across different phases of our training pipeline. In each evaluation, users are shown two relit images and asked to select the one with better lighting alignment, identity preservation, and lighting realism. While Latent-Intrinsics preserves identity better, it struggles with lighting realism and alignment. Our method achieves significantly better lighting realism and alignment compared to LumiNet, with the third-stage refinement yielding further improvements—highlighting the effectiveness of our approach.

| | Human evaluation on in-the-wild relighting ↑ | | | Human evaluation on different stages ↑ | | | |
|---|---|---|---|---|---|---|---|
| | Latent-Intrinsics | LumiNet | Ours | LumiNet | Stage I | Stage I&II | All Stage |
| Lighting alignment | 0.125 | 0.415 | **0.931** | 0.444 | 0.203 | 0.210 | **0.750** |
| Identity preservation | **0.679** | 0.050 | 0.628 | 0.050 | 0.453 | 0.677 | **0.960** |
| Lighting realism | 0.283 | 0.752 | **0.900** | 0.250 | 0.350 | 0.677 | **0.889** |

**Semantic Alignment in Relighting.** To evaluate the impact of our semantic augmentation on material-aware relighting, we group per-pixel material labels from the MIIW dataset into five physically-grounded categories: *Diffuse*, *Glossy*, *Specular*, *Metallic*, and *Uncertain/Mixed*. We then compute relighting metrics (SSIM, PSNR, RMSE) within these material-specific regions.

Table 4 presents relighting performance grouped by surface reflectance properties. Notably, the use of augmented semantic intrinsic features in Stage I yields consistent improvements across all material types, with particularly strong gains (+6%) on non-diffuse surfaces such as *Glossy*, *Metallic*, and *Specular*, which typically demand a deeper semantic understanding. Also note that all scores are averaged over the semantic mask corresponding to the relit object regions; because specular and translucent areas occupy only a small fraction of this mask, the improvements reported below likely *understate* our method's visual gains on those challenging pixels.

**Note.** Standard full-reference metrics (PSNR/RMSE/SSIM) correlate weakly with perceived lighting quality and tend to emphasize low-frequency tone over directional effects (Giroux et al., 2024; Xing et al., 2025). We therefore use them (i) for completeness on standard protocols (Tabs. 1–2) and (ii) as internal sanity checks in controlled ablations (Tab. 3); conclusions about relighting fidelity are drawn primarily from cross-scene generalization and qualitative, physically plausible evidence. Developing better evaluation metrics for lighting is an open problem and outside the scope of this work.

**Ablation: Multi-Stage Training.** Table 4 shows that our model improves consistently across material categories after Stage II. While Stage III (self-refinement) results in a slight decline in quantitative metrics on the MIIW benchmark, we observe noticeably better relighting quality on diverse in-the-

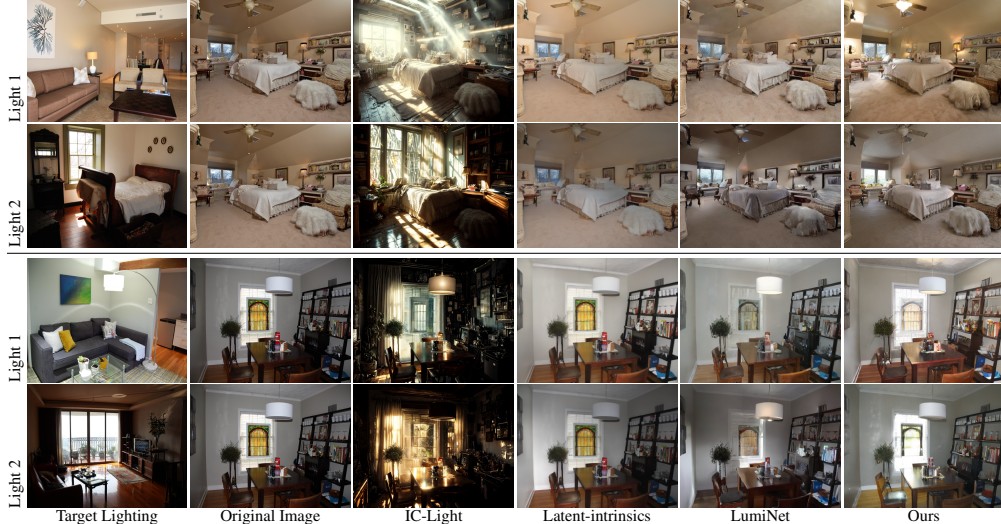

Figure 4: **Relighting comparison across two real-world images**, each shown under two target illuminations (Light 1: lamp-dominated; Light 2: sunlight through windows). IC-Light (Zhang et al., 2025a) produces stylized results with exaggerated glow and artifacts that diverge from the targets. Latent-intrinsics (Zhang et al., 2024a) captures some variation but yields low-contrast, flattened illumination with weak directionality. LumiNet (Xing et al., 2025) better matches global tone but remains overly diffuse, often missing dominant light sources and underestimating cast shadows and highlight localization. **Ours** preserves material detail and transfers both global and directional lighting, producing images that most closely match the targets lighting.

Table 4: **Performance comparison across different material semantic labels.** Background color represents relative improvement within each metric, redder (warmer) shades indicating greater enhancement compared to the baseline LumiNet, which utilizes latent intrinsic features as intrinsic representation. Large gains are observed in complex metallic, glossy or specular surfaces. Note because scene/protocol/metric are fixed for this evaluation, PSNR/SSIM/RMSE serve as internal proxies: the monotonic improvements from Stage I→II→All-Stage confirm better photometric calibration without regressions; perceptual lighting gains are corroborated by the visual comparisons and directional cues.

| Semantic Label | LumiNet Xing et al. (2025) | | | Ours (Stage I Only) | | | Ours (Stage I&II) | | | Ours (All Stage) | | |
| --- | --- | --- | --- | --- | --- | --- | --- | --- | --- | --- | --- | --- |
| | SSIM↑ | PSNR↑ | RMSE↓ | SSIM↑ | PSNR↑ | RMSE↓ | SSIM↑ | PSNR↑ | RMSE↓ | SSIM↑ | PSNR↑ | RMSE↓ |
| Uncertain | 0.6082 | 17.8499 | 0.1426 | 0.6178 | 18.0678 | 0.1373 | 0.6518 | 18.6282 | 0.1307 | 0.6460 | 18.4975 | 0.1310 |
| Diffuse | 0.5281 | 18.0883 | 0.1369 | 0.5464 | 18.3033 | 0.1320 | 0.5798 | 18.7476 | 0.1261 | 0.5731 | 18.6726 | 0.1269 |
| Glossy | 0.5720 | 18.6291 | 0.1296 | 0.5988 | 19.0248 | 0.1217 | 0.6291 | 19.6906 | 0.1152 | 0.6196 | 19.5275 | 0.1157 |
| Metallic | 0.4164 | 15.4926 | 0.1759 | 0.4608 | 16.0831 | 0.1636 | 0.4855 | 16.3285 | 0.1588 | 0.4822 | 16.3827 | 0.1581 |
| Specular | 0.3778 | 17.0860 | 0.1490 | 0.4120 | 17.6720 | 0.1394 | 0.4423 | 17.9798 | 0.1357 | 0.4365 | 18.0688 | 0.1330 |

wild scenes (Fig. 5). All scores are averaged over semantic masks covering relit object regions; because specular and translucent materials occupy only a small fraction of those regions, the metrics may *understate* our model's improvements on these challenging surfaces. Nevertheless, our method yields consistent gains across metrics, indicating that the enhanced latent representation generalizes well across reflectance types.

**Lighting Interpolation.** Our **ALI** representation is physically meaningful, capturing meaningful separation between scene-invariant properties (intrinsics) and illumination conditions (extrinsics). Fig. S.2 and Fig. 4 demonstrate relighting results where the intrinsic representation is held fixed while varying the extrinsic lighting input. Furthermore, Fig. 6 shows that, within the same scene (i.e., fixed intrinsics), interpolating or perturbing the extrinsic code produces physically plausible and continuous lighting transitions.

**Vision Backbone.** Table 5 compares different visual representations for semantic augmentation. RADIOv2.5 and MAE provide larger gains than CLIP, DINOv2/3, which leads to two observations:

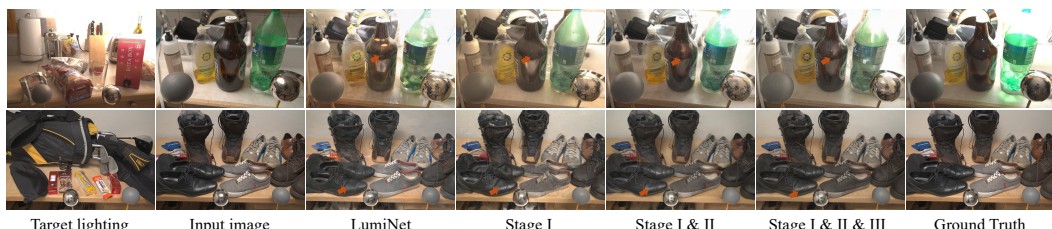

Target lighting    Input image    LumiNet    Stage I    Stage I & II    Stage I & II & III    Ground Truth

(a) Multi-stage ablation on MIIW Murmann et al. (2019) dataset.

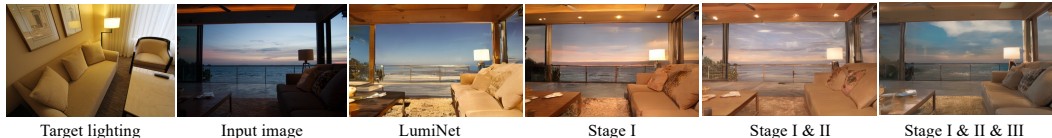

Target lighting    Input image    LumiNet    Stage I    Stage I & II    Stage I & II & III

(b) Multi-stage ablation on in-the-wild image (with bypass decoder disabled).

Figure 5: **Multi-stage ablation.** *Top*: Compared to LumiNet, our Stage I improves fine geometry details. Adding Stage II sharpens directional cues and specular effects, while the full pipeline (Stage I&II&III) produces the closest match to ground truth, with accurate shadows, highlights, and material fidelity. This progression illustrates how each stage contributes complementary improvements, consistent with the quantitative gains in Tab. 4. *Bottom:* LumiNet produces flat illumination with weak lamp cues. Stage I introduces coarse global tone but has color shift, Stage II suppresses these effects, and the full pipeline (Stage I&II&III) yields the most faithful transfer: interior warmth from the lamp is preserved while maintaining the outdoor scene, closely matching the target lighting. This demonstrates that our stage-wise design generalizes to unconstrained real-world images.

MAE, though simple, aligns well with relighting because its objective is pixel reconstruction. This keeps the feature space closely tied to the image, preserving fine-grained color, shading, and reflectance cues that are crucial for illumination modeling. In contrast, contrastive methods such as DINO and CLIP are explicitly trained to be invariant to color, local contrast, and other small appearance changes, due to strong color and contrast jitter in their augmentations. Their features therefore suppress exactly the subtle specular and color variations that encode material and lighting, making them less suitable for

Table 5: **Impact of semantic features on relighting.** Augmenting latent intrinsics with RADIOv2.5 or MAE significantly improves metrics after Stage II.

| Feature | Stage | RMSE ↓ | LPIPS ↓ | PSNR ↑ | SSIM ↑ |
|---|---|---|---|---|---|
| Latent Intrinsic Zhang et al. (2024a) | I | 0.1380 | 0.2857 | 17.5763 | 0.5461 |
| | I&II | 0.1383 | 0.2844 | 17.5463 | 0.5531 |
| MAE He et al. (2022b) | I | 0.2195 | 0.4820 | 14.2381 | 0.4571 |
| | I&II | 0.1286 | 0.2554 | 17.9861 | 0.4852 |
| DINOv2 Caron et al. (2021) | I | 0.2786 | 0.3295 | 13.2439 | 0.4646 |
| | I&II | 0.1686 | 0.3253 | 15.7945 | 0.4815 |
| DINOv3 Siméoni et al. (2025) | I | 0.1794 | 0.3588 | 15.1375 | 0.4824 |
| | I&II | 0.1654 | 0.2923 | 16.1510 | 0.5299 |
| CLIP Radford et al. (2021) | I | 0.2189 | 0.4556 | 13.9671 | 0.3987 |
| | I&II | 0.1627 | 0.3153 | 16.1333 | 0.5039 |
| RADIOv2.5 Heinrich et al. (2024a) | I | 0.1312 | 0.2673 | 17.9448 | 0.5609 |
| | I&II | **0.1260** | **0.2440** | **18.3426** | **0.5958** |

high-fidelity relighting. Consistently, a concurrent work RAE (Zheng et al., 2025) shows that an MAE-based representation with a diffusion decoder achieves better reconstruction and relighting quality than a DINO-based counterpart with the same diffusion-based decoder.

RADIOv2.5 further improves relighting performance. Its multi-resolution training enforces pixel-level consistency across scales, enhancing the reconstruction of local lighting and material effects, while its multi-model distillation injects complementary high-level semantic and structural cues. This yields features that are both pixel-aligned and semantically rich—properties that are essential for relighting, which requires both low-level appearance detail and high-level scene understanding.

## 6 DISCUSSION

This work investigated what makes a visual representation "relightable" and found a surprising result: representations from top-performing semantic encoders often hinder, rather than help, high-fidelity relighting. This reveals a fundamental trade-off between semantic abstraction, which discards essential photometric detail, and physical fidelity. Our method, **Augmented Latent Intrinsics (ALI)**, addresses this by strategically fusing features from a dense, pixel-aligned encoder into a latent-intrinsic framework. Trained solely on unlabeled real-world image pairs, ALI achieves state-of-the-art performance, outperforming methods that rely on privileged inputs, especially on challenging

Light 1 ⟶ Interpolations ⟵ Light 2

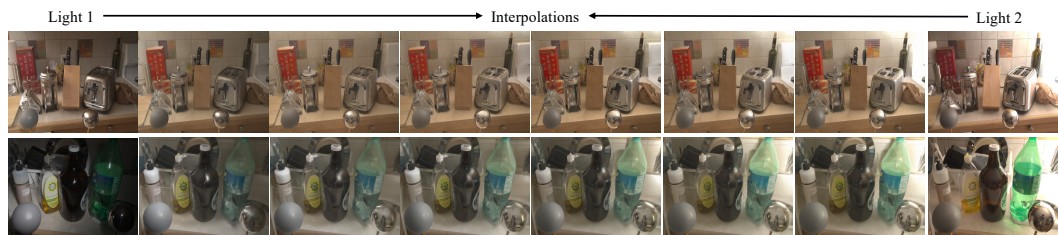

(a) Image relighting with interpolated lighting code.

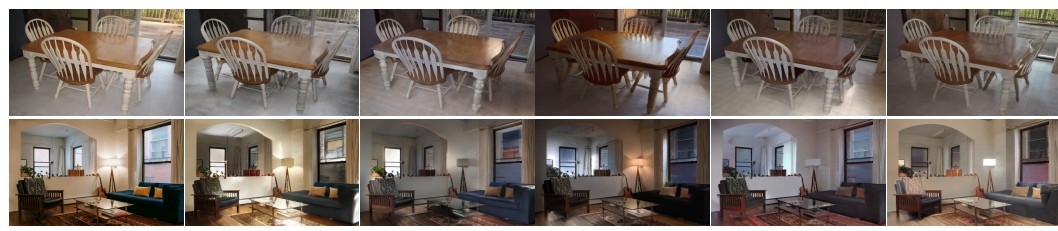

Original lighting                     Random lighting sampling

(b) Zero-Shot image relighting with unpaired, randomly sampled lighting code.

Figure 6: **Lighting interpolation and diversity. Top:** Generated images showing a smooth interpolation between two lighting codes. Note the plausible evolution of directional lighting, including the progressive appearance of sharp specular highlights on the toaster and caustic effects from the bottle. **Bottom:** In-the-wild relighting results using lighting codes sampled from random, unpaired images. Our method produces a diverse range of distinct illumination effects, plausible altering the glossy reflections on the dining table and the ambient lighting in the living room.

materials. Our findings demonstrate that the path to physically-grounded generative models lies not in simply scaling generic encoders, but in the principled analysis and targeted fusion of complementary visual priors.

**What makes a representation relightable?** Relighting demands both low-level appearance cues and high-level semantics, but most visual features entangle illumination with scene intrinsics. Our study shows that simply making features denser (e.g., DINOv2 → DINOv3) is not enough: the training objective is crucial. Pixel-reconstruction objectives such as MAE produce pixel-aligned features that preserve local details of pixel space, already outperforming contrastive methods like DINO and CLIP for relighting. RADIOv2.5 further improves relighting by combining multi-resolution training, which reinforces pixel-level consistency across scales, with multi-model distillation, which injects complementary high-level semantics. As a result, RADIOv2.5 yields a representation that is both pixel-faithful and semantically rich—exactly what relighting requires.

Our approach belongs to a growing category of *prior-driven* methods. We see the field as having two valuable pathways: (1) *Scale-heavy approaches*, which train on large synthetic corpora with full supervision (Zeng et al., 2024b; Liang et al., 2025; He et al., 2025), and (2) *Prior-driven approaches*, like ours, that learn from smaller, real-world datasets by leveraging strong inductive biases from pretrained models. We believe our work shows that thoughtful feature fusion, rather than dataset scale alone, is a powerful and accessible path toward physically coherent relighting.

The success of this fusion stems from its role as a lightweight augmenter. By injecting frozen features from a well-formed visual prior, we stabilize early optimization and gently guide the encoder toward a semantics-aware solution. Similar stabilization effects have been observed in image synthesis (Zhang et al., 2025b; Yu et al., 2025). Our results complement those findings by showing that these visual priors may also carry useful *physical* information crucial for relighting.

**Limitations and Future Work.** Despite improved material fidelity, our generative model can still shift fine details and occasionally confuses color with lighting—an inherent ambiguity in inverse rendering. Shadows are often plausible but are guided by learned priors rather than explicit geometry, and thus may fail in uncommon configurations. While we tested our fusion approach on a single backbone (LumiNet), we believe the principle should transfer to other inverse-rendering encoders with minimal change, representing a promising direction for future work.

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

## A   IMPLEMENTATION DETAILS

### A.1   TRAINING

All experiments are conducted on a node equipped with 8 NVIDIA A6000 Ada 48GB GPUs. The model is trained at a resolution of $512 \times 512$ with an effective batch size of 64 (including gradient accumulation).

For Stage I, we employ a scene-aware sampler that ensures each batch contains images from the same scene captured under different lighting conditions. For Stage II and Stage III, we use standard random sampling across scenes. Additionally, in Stage III, we include a small proportion (10% of the training set) of identity relighting samples—where the input and target share the same lighting—to explicitly enforce geometry preservation.

Stage I is trained for 4 epochs, while Stage II and Stage III are trained for 2 epochs each. Due to differences in model size and parameter updates, Stage I training takes approximately 8 hours on a single GPU, while each of Stage II and III takes around 10 hours. We use the AdamW optimizer for all training stages.

### A.2   LIGHTING ZOO GENERATION

To enhance the model's generalization to real-world inputs, we use the Stage II checkpoint to generate a large-scale in-the-wild relighting dataset, referred to as the *Lighting Zoo*. We curate approximately 6,000 diverse images from open-source datasets including IIW (Bell et al., 2014), RealEstate-10K (Zhou et al., 2018), and DL3DV (Ling et al., 2024). Fig. S.1 shows a selection of pseudo-relit pairs; note the plausibility and diversity of the synthesized lighting.

During generation, images are randomly grouped into batches. Each original image serves as the input, while the target lighting condition is randomly sampled from other images within the same batch. For each scene, we generate seven relit versions under different lighting conditions. The process is distributed across multiple NVIDIA A6000 Ada 48GB GPUs for efficiency. In total, generating the Lighting Zoo requires approximately 500 GPU hours.

## B   SEMANTIC ALIGNMENT IN RELIGHTING

### B.1   SEMANTIC GROUP CURATION

The MIIW dataset Murmann et al. (2019) provides detailed semantic annotations, including binary masks for 31 distinct material labels. To efficiently evaluate relighting performance across materials with varying reflectance properties, we regroup these fine-grained labels into five broader categories: *Diffuse*, *Glossy*, *Specular*, *Metallic*, and *Uncertain/Mixed*.

To ensure consistency and physical relevance, we utilize advanced reasoning model GPT-O3 with prompt - "cluster those class label into a higher-level BRDF-style categories." - to assist in the semantic reasoning and grouping process. The full mapping between original material labels and grouped categories is provided in Table S.1.

## C   MORE VISUAL RESULTS

We present additional visual results on the MIIW dataset and in-the-wild images. Fig. S.2 illustrates the same scene relit under different target lighting conditions. Fig. S.3 demonstrates relighting on in-the-wild examples.

Benefiting from the proposed augmented latent-intrinsics feature, our method is capable of relighting a wide variety of scenes while preserving geometry and scene identity. These include: 1) a real indoor image captured by a phone, 2) a painting, and 3) an internet-sourced image.

Table S.1: **Mapping of material classes to reflectance clusters.** Each material class is assigned a cluster label based on its dominant reflectance properties: Diffuse, Glossy, Specular, Metallic, or Uncertain. The cluster name is shown only at the start of each group for clarity.

| Cluster | Class Index | Class Label |
|---|---|---|
| Diffuse | 3 | Cardboard |
| | 5 | Concrete |
| | 6 | Cork/corkboard |
| | 7 | Dirt |
| | 8 | Fabric/cloth |
| | 9 | Foliage |
| | 10 | Food |
| | 11 | Fur |
| | 14 | Laminate |
| | 16 | Linoleum |
| | 21 | Paper/tissue |
| | 25 | Sponge |
| | 26 | Styrofoam |
| | 29 | Wallpaper |
| | 31 | Wicker |
| | 32 | Wood |
| | 33 | Stone |
| | 34 | Chalkboard/blackboard |
| | 35 | Carpet/rug |
| | 36 | Brick |
| Glossy | 4 | Ceramic |
| | 15 | Leather |
| | 23 | Plastic — opaque |
| | 24 | Rubber/latex |
| | 27 | Tile |
| | 30 | Wax |
| Specular | 12 | Glass |
| | 18 | Mirror |
| | 22 | Plastic - clear |
| Metallic | 17 | Metal |
| Uncertain / Mixed | 0 | unassigned |
| | 1 | I can't tell |
| | 2 | More than one material |
| | 13 | Granite/marble |
| | 19 | Not on list |
| | 20 | Painted |
| | 27 | Tile |
| | 28 | splitshape |
| | 37 | Skin |
| | 38 | Water |
| | 39 | Hair |
| | 40 | no_consensus |

# D  USE OF LLM

We use large language models for latex help, and help correct our grammar.

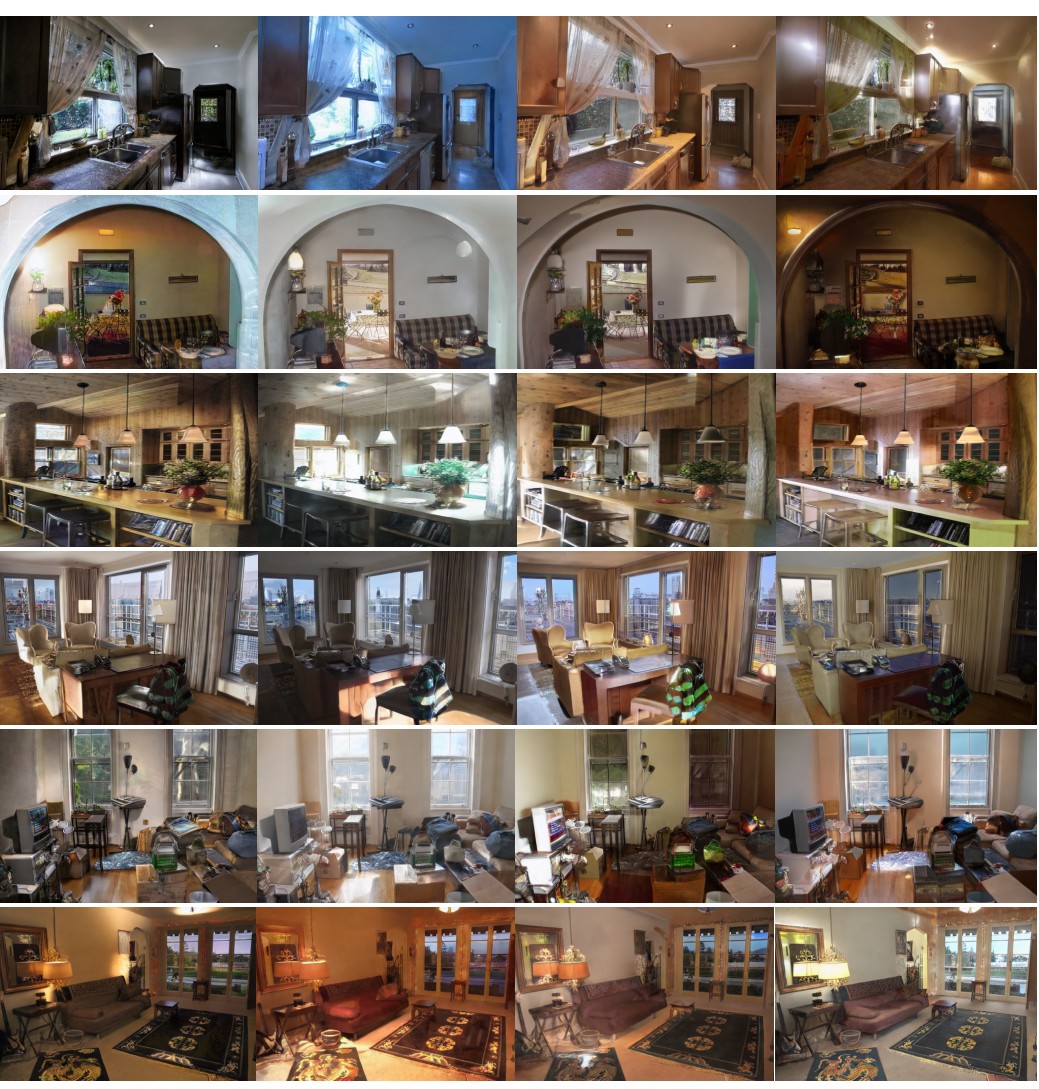

Figure S.1: **Lighting Zoo.** Pseudo-relit image pairs generated by our method for the third-stage refinement.

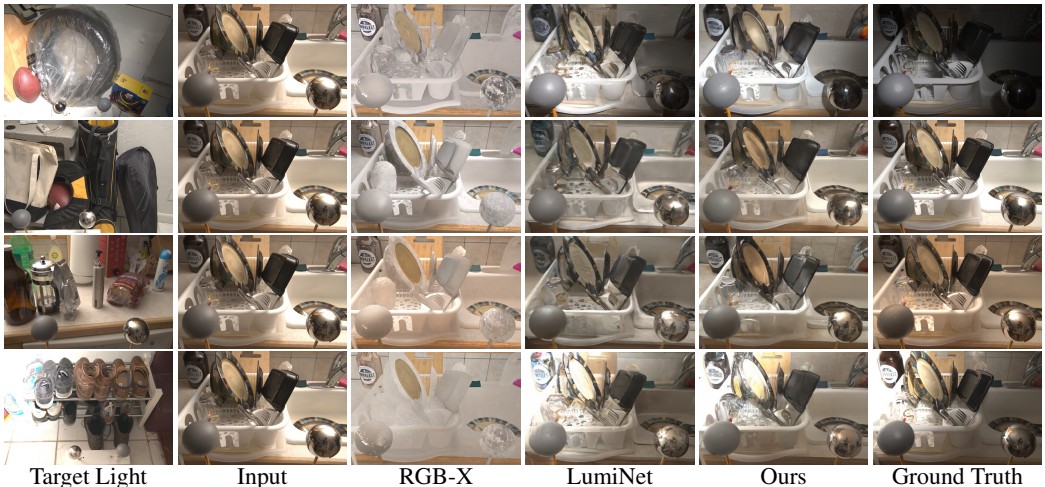

| Target Light | Input | RGB-X | LumiNet | Ours | Ground Truth |

Figure S.2: **Scene relighting comparison on the MIIW Murmann et al. (2019) dataset.** Each row corresponds to a different target lighting condition (left). We compare the relighting outputs of RGB-X, LumiNet, and our method against the ground truth (with bypass decoder disabled).

| Input | Relighting 1 | Relighting 2 |

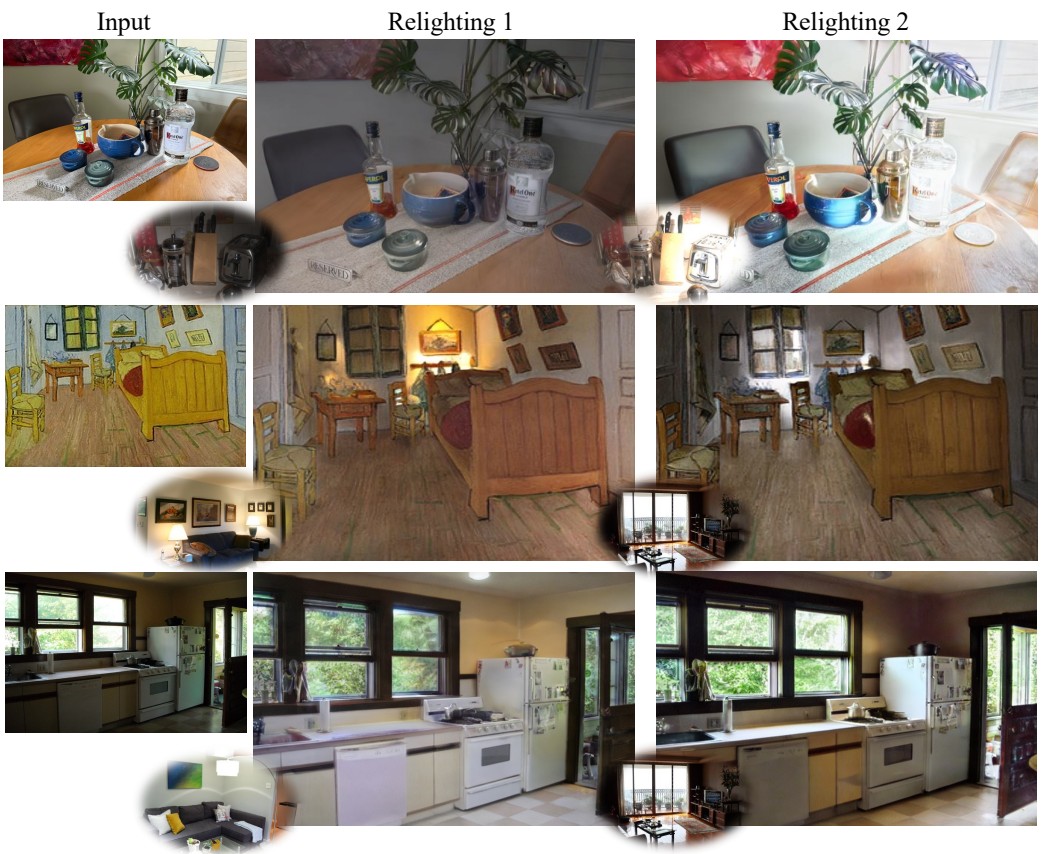

Figure S.3: **In-the-wild relighting.** Our method is capable of relighting a wide variety of scenes while preserving geometry and scene identity. These include: 1) a real indoor image captured by a phone, 2) a painting, and 3) an internet-sourced image. The target lightings are indicated in the bottom left corner of each image.

