# OpenReview forum: "What Makes a Representation Relightable? Probing Visual Priors via Augmented Latent Intrinsics"
_ICLR.cc/2026/Conference — Submitted to ICLR 2026_

### Official Review · Reviewer_MbbU · 2025-10-31

**Soundness:** 2
**Presentation:** 2
**Contribution:** 3
**Rating:** 6
**Confidence:** 4

**Summary:**

The paper builds upon LumiNet (Xing et al., 2025\) and Latent Intrinsics (Zhang et al., 2024\) for image-to-image relighting. The proposed method incorporates an additional semantic encoder (a vision foundation model) to augment the extracted latent intrinsics. The paper is framed within the context of how visual priors from foundation models help the relighting performance and there is some discussion on which vision foundation models are good for the relighting task. The method is trained solely on image pairs without the need for privileged information such as depth or GT lighting and the training protocol includes an additional self refinement step. The method is evaluated on the MIIW dataset and ablations show the performance on different material sub-groups and the interpretability of the latent extrinsic.

**Strengths:**

* The paper introduces semantic context from a vision foundation model (semantic encoder) for the task of image-to-image relighting. The proposed method improves upon the performance of prior works trained on diverse datasets, such as LumiNet, over the MIIW dataset.
* The authors show that the latent intrinsics and extrinsic are well trained and in particular show that interpolating between two extrinsics still produces plausible lighting.
* The authors highlight specific limitations of the metrics (PSNR, SSIM, RMSE) and made clear that this is still an open research problem to properly evaluate the improvements for relighting specific challenging material classes.

**Weaknesses:**

* The central claim of the paper on “A controlled protocol and operational criteria for probing relightability, explaining why popular semantic encoders fail despite strong recognition performance.” could be better explained. Figure 1 (right) shows a negative correlation between ImageNet 1K linear probing accuracy and relighting performance for pretrained vision foundation models. The authors also touch on the choice of vision encoder in Section 5 and Table 4\. Perhaps, the authors could share more details on the experimental settings for Figure 1 (right) and include discussions on why the proposed method, based on RADIOv2.5, overcomes the limitations. Another suggestion would be to compare performance with the different model sizes for the different semantic encoders (e.g. RADIOv2.5B, RADIOv2.5-L, RADIOv2.5-H).
* The paper relies primarily on qualitative examples to demonstrate the real-world generalizability of the proposed approach, and also the need for Stage III in the training pipeline. LumiNet, following the practice of LightIt (Kocsis,et al., 2024), conducted a user study to assess the performance across three metrics: image quality (I-PQ), lighting quality (L-PQ) and prompt alignment (P-PQ). A similar study could strengthen the claims of the paper.
* It is not clear to this reviewer how the proposed approach (ALI), where features from a dense, pixel-aligned encoder are “strategically” fused (line 464). The fusing occurs in Stage 1 of the training, where N (what is the number of N) feature maps are selected from $E_{sem}$, upsampled and concatenated to form H, followed by a learnable P\_theta layer. Details of N, dimensions of $P_\\theta$ and other fusion details would make this paper more convincing in terms of the insights it yields. This will help the reader appreciate the main contribution of this so-called light weight fusion adapter (line 97\)

**Comments**:

* MAE paper repeated twice in the references
* Define MIIW early (line 91\)
* Please put in the definitions of E\_theta, E\_sem etc. into Fig. 2, else it is really hard to follow.
* Fig. 4 – comparisons with IC-Light, this work was not mentioned in the literature survey, do add it in so readers can appreciate the comparison better
* Table 1: which is the raw and which is the color-corrected?

**References**:
Kocsis, P., Philip, J., Sunkavalli, K., Nießner, M., & Hold-Geoffroy, Y. (2024). Lightit: Illumination modeling and control for diffusion models. In *Proceedings of the IEEE/CVF Conference on Computer Vision and Pattern Recognition* (pp. 9359-9369).

Xing, X., Groh, K., Karaoglu, S., Gevers, T., & Bhattad, A. (2025). Luminet: Latent intrinsics meets diffusion models for indoor scene relighting. In *Proceedings of the Computer Vision and Pattern Recognition Conference* (pp. 442-452).

Zhang, X., Gao, W., Jain, S., Maire, M., Forsyth, D., & Bhattad, A. (2024). Latent intrinsics emerge from training to relight. *Advances in Neural Information Processing Systems*, *37*, 96775-96796.

**Questions:**

* How does the computational complexity/runtime of the proposed method compare to existing methods such as LumiNet and Latent Intrinsics
* Do any specific aspects of the unique training strategies for RADIOv2.5 contribute to the improved performance?
* Is the relighting encoder, E\_theta consists of the “Intrinsic encoder” and “Extrinsic encoder”? (fig. 2). If so, where is the projection layer $P_\\theta$ in fig 2.? From the VRM (“$E_{sem}$”?).
* How is the training of the “Extrinsic encoder” done as shown in Fig. 2? It is not clear from the description of Stage 1 in sec. 4 how it was involved.
* In fig 2, how are the “latent-extrinsics” trained/frozen? Can elaborate better with the description in stage 1 or stage 2 (or both?)
* Table 1: is it possible to train ALI with just MIIW train-set only so it is comparable with the upper portion, also the bold results are confusing, or is this an error?
* (Suggestion for improvement). Table 3 can include  latent-intrinsics as well, so that the improvements and comparisons for the difficult materials are even more convincing.

---

> ### Author Response · Authors · 2025-11-26
>
> Thank you for your insightful feedback!
>
> **Q1. Clarification on Fig. 1.**
>
> The ImageNet-1K Top-1 accuracy refers to results obtained via linear probing on the ImageNet-1K dataset, as reported in the respective original papers.
>
> **Q2. Discussion on why our ALI (w. RADIOv2.5) achieves the best performance.**
>
> Thanks for the insightful question. As mentioned in the introduction, relighting is a task that requires both low-level and high-level information. Relighting inherently requires both low-level and high-level information, as stated in the introduction. Our ALI (with RADIOv2.5) consistently outperforms competing methods, which directly supports this claim. Specifically, the dense pixel-level representation learned by RADIOv2.5, combined with multi-teacher distillation, provides features that are both finely aligned with image appearance and semantically rich. This combination is precisely what benefits the relighting task and explains the observed performance gains.
>
> **Q3. The different sizes of RADIO features.**
>
> Thanks for the suggestion, we now show the results with a different size of the encoder. As we see from the following table, 2.5-h outperforms the 2.5b and 2.5l models. We chose RADIOv2.5-h because it achieves the best performance among them and is distilled from a larger set of teacher models (including Florence-2).
>
> +---------+---------+---------+---------+----------+---------+
>
> | Encoder | Stage   |  RMSE   | LPIPS   |  PSNR    |  SSIM   |
>
> +---------+---------+---------+---------+----------+---------+
>
> | 2.5-b   | I & II  | 0.1288  | 0.2488  | 18.2346  | 0.5898  |
>
> | 2.5-l   | I & II  | 0.1281  | 0.2464  | 18.2571  | 0.5901  |
>
> | 2.5-h   | I & II  | 0.1260  | 0.2440  | 18.3426  | 0.5958  |
>
> +---------+---------+---------+---------+----------+---------+
>
> **Q4. User study.**
>
> Thank you for the suggestion. We conducted a user study following evaluation protocols similar to those used in the LumiNet and LightIt papers. The updated results are presented in our revised manuscript (see Table 4). In summary, our ALI model, when combined with RADIO features, demonstrates a significantly stronger user preference compared to other baseline methods.
>
> **Q5. The computation add-on.**
>
> At the same output resolution (512×512), our method introduces approximately a 15% increase in additional runtime compared to LumiNet, and incurs roughly 2× the add-on time relative to Latent Intrinsics. However, it is important to note that this comparison is not entirely symmetric, as Latent Intrinsics does not employ diffusion models, which inherently contribute to increased computational cost.
>
> **Q6. Clarification on Fig. 2.**
>
> Yes, the encoder (E_\theta) contains the latent-extrinsics encoder as well. The projection layer is shown as the orange bar after the VRM pattern. We added the notation in the updated figures to make it clearer.
>
> **Q7. Clarification on latent-extrinsics training/frozen.**
>
> The extrinsic encoder is trained by applying the hyperspherical regularization loss. We updated this part in the paper (please see Lines 208–234).
>
> **Q8. Training ALI on MIIW only, wrong bold results.**
>
> Thank you for the suggestion. Training ALI on MIIW only involves training the decoder, similar to how LumiNet is trained solely on MIIW. However, conducting this comparison is beyond the scope of this paper. We have also corrected the bolded result in the table—thank you for bringing this to our attention.
>
> **Q9. The Latent Intrinsics result in Table 3.**
>
> Thank you for the suggestion. We have included the corresponding results and plan to integrate them into the camera-ready version, as the current table is already quite dense and will be reorganized accordingly. We agree that this addition provides important context. As expected, the Latent Intrinsics model—trained solely on the MIT dataset and without diffusion—achieves stronger quantitative performance overall. However, it performs poorly in qualitative evaluations, as illustrated in Figures 3 and 4. These results indicate two key limitations: (1) the model tends to overfit to the training distribution, and (2) it exhibits limited generalization capability—neither of which is captured by existing metrics. As noted in Lines 369–374, developing more robust and perceptually aligned evaluation metrics remains an open research challenge and is beyond the scope of this work.
>
> +-----------+---------+---------+---------+
>
> | Material  |  SSIM   |  PSNR   |  RMSE   |
>
> +-----------+---------+---------+---------+
>
> | Uncertain | 0.7720  | 20.239  | 0.1118  |
>
> | Diffuse   | 0.7307  | 20.530  | 0.1062  |
>
> | Metallic  | 0.6536  | 17.887  | 0.1344  |
>
> | Glossy    | 0.7345  | 20.907  | 0.1010  |
>
> | Specular  | 0.6472  | 20.333  | 0.1090  |
>
> +-----------+---------+---------+---------+

---

### Official Review · Reviewer_oftv · 2025-10-31

**Soundness:** 3
**Presentation:** 4
**Contribution:** 2
**Rating:** 4
**Confidence:** 3

**Summary:**

The paper tries to answer the question of what makes a representation relightable? They show interesting trade-off between global and local contexts for relighting task and clasification task. They propose Augmented Latent Intrinsics (ALI) - a fusion adapter that injects semantic information from frozen priors into the latent intrinsic relighting pipeline. They propose a three-state training scheme that first computes the ALI featires, then retrains the diffusion decoder and finally refines the predictions through a self-supervised approach. Despite training on unlabelled real image pairs, the model predicts more accurate relighting on specular and glossy surfaces as compared to prior work. They show the benefit of their proposed approach through various ablations studies. However, the metrics are not better than prior works. Qualitative results have some big artifacts (see weaknesses) and multi-stage pipeline does not seem justified. Further, several key details are missing and some need more explanation (see weaknesses and questions).

**Strengths:**

- The paper is well motivated. The authors try to answer the question - what makes a feature relightable. They explain the trade-off between semantic abstraction and photometric fidelity for relighting tasks. Fig 1 is very interesting.
- Leverage powerful visual priors and show the benefit of semantic feature injection towards improved image relighting performance. The features also enable generalization on imagenet classification and relighting. Most prior work generalizes to one of the two tasks, but not both.
- Nice practical idea to generate pseudo pairwise images in stage 3 of training to overcome lack of pairwise real-image dataset.
- Despite being trained only on same scene relighting, the model generalizes to relighting across different scenes.
- The method achieves very good relighting on images with specular and reflective surfaces, improving upon other methods.
- Ablation studies show the benefit of the multi-stage training pipeline and the proposed ALI features.

**Weaknesses:**

- A key motivation of the paper is Fig 1. But there is no explanation on how the models were trained to obtain the metrics in Fig 1. Do you replace the VRM module in your pipeline with CLIP or DINO modules? How was the model trained to obtain metrics in table 4? Similarly, how was the imagenet classification model trained? Were the features passed through a few additional layers or directly to a softmax layer to get class labels?
- Stage 3 training does not seem to help much. In Fig 5a, there is limited benefit to stage 3 as compared to stage 2. Fig 5b shows several artifacts in the cloud regions in stage 3 as compared to stage 2. From Fig S.1, it is seen that for the same scene, there are several artifacts across the different images in the lighting zoo. This design choice could be reason for artifacts in the predicted image.
- Improvement over latent intrinsics is unclear. In tables 1 and 2, latent intrinsics has better metrics. (The note about metrics not capturing specular highlights is true and it is acknowledged) Fig 3 shows the improvement of the proposed method over latent intrinsics, especially on specular surfaces. However, in Fig 4, the proposed method introduces significant artifacts (rows 3,4), affects the photorealism of the generated image. I would prefer the trade-off in results from latent intrinsics since maintaining scene consistency is vital in image relighting applications. Further, the authors state that latent intrinsics is trained only on MIIW dataset, which consists of narrow FOV images. Despite this, latent intrinsics has better performance on wider FOV images in Fig 4.
- Authors state that the model improves upon the state of the art model performance on the MIIW benchmark. But in tables 1 and 2, the performance is worse than latent intrinsics. And both methods do not use any privileged label information. So this claim is overstated. The paper should clarify the context of the results more carefully.
- Despite training on both MIIW and big time dataset, the authors evaluate the models only on MIIW dataset. Even if direct comparisons to other methods are not possible, evaluating the generalization of the paper across different test datasets would strengthen the paper.
- Cross-dataset generalization was not evaluated. For example, the generalization of a model trained on MIIW dataset but evaluated on big time dataset was not conducted. This can further strengthen the claims about the benefits of ALI feature representation.
- In Fig 5b, stage 2 and 3 images have a dark shadow on the floor area near the lamp. This is not photorealistic. LumiNet result is bright in that area and that is more photorealistic.
- The lighting zoo is an important aspect of the paper. So it should be included in the main paper rather than the appendix.

**Questions:**

- Does Fig 1 show the performance of different encoders used instead of the VRM encoder? If not, how much of the model's performance can be attributed to the feature contribution vs pipeline design.
- During stage 3 training, was the model initialized with stage 2 weights? If yes, this could lead to catastrophic forgetting. Given the input image in stage 3 is from the stage 2 predictions, the stage 3 model could be compensating for the mistakes in the predicted image. This might be the cause for artifacts in stage 3 predictions (see Fig 5b). Fig S.1 has several artifacts in across images of the same scene. It is observed in rows 1 (banana and bottles near the window pane), row 4 (jacktes and building outside the window), row 5 (buildings outside and carry bag), etc. Maybe a weighted average of stage 2 and stage 3 weights might help balance the model better.
- Eq 2 formulation seems a bit too simplistic. Adding is sensitive to scale of the features and does not extract meaningful information between the features. Were alternate methods explored to combine the features? Some of them are - learnable weighted addition, FiLM layer or cross attention layer.
- During stage 3 training, the paper states that the network is trained to ignore artifacts and focus in essential properties. How is this achieved?
- Why does stage 3 performance not improve on MIIW dataset? Is it because the dataset is too simplistic?
- In table 1, SA-AE has best metrics for the first two columns. It is indicated wrongly in the paper.
- In Fig 5b, is the target light colour yellow or white? It is difficult to determine from the image and hence, it is hard to say if lumiNet prediction is more accurate or your prediction is more accurate.
- An interesting result: In fig 4, the ceiling light is ON is one case and OFF in the other case depending on ambient light of the target image in that area. How is the modelling learning this? It is quite interesting.
- Despite using bypass decoder, the text on the bottles in the generated image is not reconstructed properly (Fig S2, S3). Why is that? While the diffusion models struggle on text reconstruction due to VAE compression, one would expect that the bypass decoder would address it.

---

> ### Author Response · Authors · 2025-11-26
> **Reponse 1/2**
>
> Thank you for the detailed and insightful feedback!
>
> **Q1. Clarification on Fig. 1, Table 4 regarding how the model was trained.**
>
> Thank you for the question. As described in Lines 196–201, we extract a hierarchy of feature maps from the VRM encoder, specifically from the outputs of its intermediate layers. For consistency, all methods reported in Table 4 are trained on the same dataset, using identical hyperparameters and the same number of training iterations. Regarding the ImageNet classification results, we adopt the ImageNet-1K linear evaluation protocol as reported in each method’s original paper. This involves training a single linear classifier on top of frozen features and reporting its top-1 accuracy on the ImageNet-1K benchmark. We have clarified this detail in the main paper for improved transparency.
>
> **Q2. Concerns about Stage 3 training.**
>
> As described in the paper (Lines 237–239; 819–821), Stage 3 introduces a self-refinement mechanism aimed at improving generalization to in-the-wild internet images, particularly by enhancing the recovery of fine-grained details in relit outputs. As shown in Fig. 5b, this stage allows the model to more accurately reconstruct textures, such as the carpet and the pillows on the sofa. The effectiveness of Stage 3 is further validated by our user study, which demonstrates significant improvements over Stage 2: a +0.540 increase in win-rate for lighting alignment, and additional gains of +0.287 in identity preservation and +0.212 in lighting realism. Compared to LumiNet, our approach achieves even more pronounced gains—+0.516 in lighting alignment, +0.158 in lighting realism, and a substantial improvement in identity preservation (from 0.050 to 0.628)—while simultaneously boosting both perceptual quality and lighting fidelity.
>
> **Q3. Question about the improvement against Latent Intrinsics.**
>
> One of the primary limitations of Latent Intrinsics is its inability to accurately transfer target lighting, often resulting in only localized changes. For instance, in rows 2 and 4, it fails to increase the intensity of exterior illumination, while in rows 1 and 3, it does not activate the interior lights specified by the target lighting. Additionally, structural artifacts are observed—for example, in row 1, the ceiling appears erroneously “split” into two segments. Our user study further supports these findings, demonstrating a higher win rate for our method compared to Latent Intrinsics. Moreover, the features used in Latent Intrinsics (which also serve as the primary input for LumiNet) struggle to differentiate complex material properties, as corroborated by the quantitative results in Table 4.
>
> **Q4. Clarification on the MIIW benchmark.**
>
> Thank you for the suggestion. As clarified in the revised manuscript (Lines 301–312), our method demonstrates further improvements in quantitative performance compared to the state-of-the-art diffusion-based approach.
>
> **Q5. Concerns about Stages 2 and 3 against LumiNet (Fig. 5b).**
>
> We acknowledge the reviewer's observation regarding identity preservation in LumiNet—for example, the inconsistencies in the pillow on the sofa, the carpet, and table details. While we partially agree with this assessment, it is important to recognize that such qualitative judgments can be inherently subjective. To provide a more objective perspective, we conducted an extensive user study (see revised Table 4), which involved multiple participants and consistently demonstrated that both Stage 2 and Stage 3 contribute significantly to improved performance. Specifically, identity preservation improves from 0.677 to 0.960, and lighting realism from 0.677 to 0.889. Furthermore, Stage 3 delivers a substantial boost in lighting alignment for in-the-wild scenes, increasing from 0.210 to 0.750.
>
> **Q6. Lighting Zoo should be in the main paper.**
>
> Good point. We have highlighted the Lighting Zoo dataset in the revised version of the paper and plan to include it in the main paper for the camera-ready submission, following a broader reorganization based on reviewer feedback.
>
> **Q7. Details on Fig. 1. How much of the model's performance can be attributed to the feature contribution vs pipeline design.**
>
> The reported relighting performance is based on ALI models using different VRM feature extractors. The ImageNet-1K Top-1 accuracies are based on linear probing results as reported in the respective original papers. Since all features are integrated into ALI using the same injection strategy, we believe the observed differences in relighting performance are primarily attributable to the quality and representational strength of the underlying features.

---

> ### Author Response · Authors · 2025-11-26
> **Reponse 2/2**
>
> [Continued]
>
> **Q8. Details of the model training, why only use the sum operation?**
>
> Thank you for the insightful suggestion. We initially explored concatenating RADIOv2.5 features along the channel dimension, but this approach resulted in slightly worse performance compared to simple feature addition (PSNR: 18.28, SSIM: 0.5814, LPIPS: 0.2421, RMSE: 0.1255). As a result, we adopted feature addition in our final model. Although this analysis was not originally included in the paper, we appreciate your suggestion and have now added it to the supplementary material to benefit the broader community.
> Additionally, our method employs a learnable weighting mechanism that adaptively scales the contribution of each feature layer, effectively acting as a feature-wise learnable scalar. We also note that for the relighting task—where precise spatial alignment is critical—the difference between feature summation and channel-wise concatenation is likely to have limited impact, provided the features remain spatially coherent.
>
> **Q9. During Stage 3 training, the paper states that the network is trained to ignore artifacts and focus on essential properties. How is this achieved?**
>
> As described in the paper (Lines 234–239, 817–821), Stage 3 utilizes synthetic relit images as inputs and real images as supervision targets. This setup ensures that the model consistently learns to denoise toward clean, ground-truth targets. We hypothesize that even if the input relit images from Lighting Zoo contain moderate errors, the model can learn to suppress such artifacts during training, resulting in tolerable degradation. Additionally, we include a subset (10%) of input–target pairs where the same real image is used as both the input and the target. This encourages the model to remain robust and identity-preserving, even when the input is artifact-free. These strategies collectively contribute to the improvements observed in our user study.
>
> **Q10. Why does Stage 3 performance not improve on the MIIW dataset? Is it because the dataset is too simplistic?**
>
> Stage 3 is primarily designed to enhance relighting performance on in-the-wild images. While the MIIW dataset is not simplistic, it exhibits spatial bias—many images share similar lighting patterns due to the use of a calibrated camera and controlled lighting rig for capturing multi-illumination from different directions. Although such a setup is valuable for evaluation, it does not generalize well to real-world, unconstrained scenarios. We will clarify this point in the revised manuscript.
>
> Accordingly, the self-refinement mechanism introduced in Stage 3 specifically targets improved relighting in out-of-distribution scenes. Therefore, we apply this refinement only to real-world, scene-scale images. Our user study confirms that incorporating Stage 3 leads to clear improvements in the perceived quality of in-the-wild relighting results.
>
> **Q11. Interesting result in Fig. 4.**
>
> Thanks for noticing it. It is quite possible, but hard to pinpoint exactly. Another possibility is that there are some small ceiling lamps in the first row’s target lighting, and that might provide a signal for turning on.
>
> **Q12. Clarification on Fig. S2 and S3.**
>
> Good observation! We intentionally did not use the bypass decoder in Fig. S2 to better isolate and assess the performance gains attributable to ALI. We have updated the corresponding description in the paper for clarity.
> Regarding Fig. S3, the visual issue was caused by image compression. The input image’s resolution was very large, and subsequent bilinear downsampling exacerbated the artifact. We have corrected this in the revised paper—thank you again for pointing it out.

---

### Official Review · Reviewer_aB99 · 2025-10-31

**Soundness:** 3
**Presentation:** 3
**Contribution:** 2
**Rating:** 2
**Confidence:** 2

**Summary:**

This paper proposes Augmented Latent Intrinsics (ALI) for image relighting tasks. It is a three-stage training pipeline that is built around the prior relighting method LumiNet by freezing and training different components at different stages. The main idea is to inject semantic features from a frozen vision encoder into the intrinsics processing pathway for LumiNet. In the experiment section, relighting results are reported with comparison to IC-Light, Latent-intrinsics, and LumiNet. Additional experiment is conducted to show the effect of using different vision encoders for the semantic features.

Overall, the paper asks a very interesting research question, but the current analysis does not really provide an answer to it. The main part of the paper is more like a standard relighting paper with a minimally designed component to inject semantic features into the existing relighting framework LumiNet.

**Strengths:**

- The paper asks a very interesting question: what makes a representation relightable? This could lead to a deeper understanding of a more effective way of leveraging existing feature encoders for tasks beyond relighting.

- The analysis in Table 3 on relighting performance grouped by different material types is interesting.

**Weaknesses:**

- Unfortunately, after reading the paper, I don't think the authors actually answer the question of what makes a representation relightable. The whole paper is written in a confusing way of mixed messages: the title and introduction focus more on the semantic representation, while the actual content section reads like a regular relighting paper. The only part that echoess with the main question is in Table 4, but only the relighting performance from different encoder backbones are reported without any further insight. This analysis is lacking depth. It is still unclear to the reader what makes a representation relightable.

- Regarding the actual design for the relighting, the only novel or interesting part is in Stage I with the injection of vision encoder features into the LumiNet features. However, fusion of different features is a common practice in multimodal learning. There are not a lot of new insights here. Other designs such as the Lighting Zoo is reasonable but not significantly novel.

- Tables 1 and 2 suggest that the proposed method is slightly better than LumiNet but worse than SOTA. Combining with the above point on novelty, it is unclear if the proposed method is making a significant contribution.

**Questions:**

1. Can the authors discuss more on the answer to 'what makes a representation relightable'?

2. Can the authors discuss more about the performance of the proposed method in Tables 1 and 2?

---

> ### Author Response · Authors · 2025-11-26
>
> Thank you for your constructive feedback!
>
> **Q1: Answer to 'what makes a representation relightable**
>
> We first respectfully disagree with the reviewer and direct them to Section 1 (Page 2 L 076-085), where we state: "Features from encoders like CLIP and DINO...systematically degrade relighting performance...the very inductive biases that create powerful semantic representations appear to discard the fine-grained photometric and spatial information required for physical reasoning." This is largely our answer: Relightable representations must preserve spatial frequency content (photometric detail) over semantic abstraction. We provide in Section 1: Semantic encoders optimize for invariance, discarding illumination-relevant cues. Empirical visualization (Figure 1): CLIP/DINO produces blurry highlights; MAE/RADIO preserves sharp specularities. Quantitative validation (Table 4): In short, Dense encoders outperform semantic ones by 2+ PSNR points
>
> We have further incorporated a detailed discussion in our revised manuscript ( L457-476; L515-523). We echo the key observation as follows:
> - (1) The contrastive training objective adversely affects relighting performance. Specifically, the data augmentations used in DINO and CLIP—such as color and contrast jittering—diminish the model’s sensitivity to subtle color and local detail variations, which are essential for accurate relighting. This explains why a simpler MAE-based representation yields better performance than DINO. A concurrent work, RAE [1], independently arrives at a similar conclusion, demonstrating that MAE achieves superior reconstruction when paired with a diffusion decoder—highlighting its suitability for relighting tasks.
>
> - (2) Relighting requires both low-level and high-level information. Even though the detail matters in relighting we still need a multi-modality semantic feature to differentiate between different materials types --- this is possible through better semantic encoders and we found RADIO to have the best of both capabilities and therefore, it helps in improving both.  RADIOv2.5 addresses this by employing multi-resolution training, which enhances pixel-level alignment, and multi-model distillation, which enriches high-level semantic features. Together, these strategies enable the construction of a more comprehensive representation, leading to improved relighting quality.
>
> **Q2: Concern about the novelty.**
>
> We first want to emphasize that this paper is not about a specific method but rather a systematic investigation of which representations are useful for relighting. Our findings oppose a large body of work that straightforwardly adopts DINO or the best semantic encoder for many downstream tasks. Furthermore, our discovery that RADIO can capture both better photometric fidelity and semantic abstraction opens up interesting perspectives for the broader research community, where physically grounded tasks require a balance of semantic abstraction and reconstruction capabilities.
>
> **Q3: Discussion on Table 1 and 2.**
>
>   As discussed in L369–374, standard metrics correlate weakly with perceived lighting quality and tend to emphasize low-frequency tone over directional effects. Please also see [2] for a detailed evaluation on how these metrics poorly correlate with human judgment. To better demonstrate the advantages of our approach, we provide qualitative comparisons in Fig. 3. In addition, we have now included a user study, which clearly shows that our method yields a significant improvement over existing approaches.
>
> **Reference:**
>
> [1] Zheng, et al. Diffusion Transformers with Representation Autoencoders. Arxiv 2025.
>
> [2] Giroux, et al. Towards a perceptual evaluation framework for lighting estimation. ICCV 2023.

---

### Official Review · Reviewer_Ru4M · 2025-11-06

**Soundness:** 3
**Presentation:** 3
**Contribution:** 3
**Rating:** 6
**Confidence:** 4

**Summary:**

This paper investigates which representation is most condusive to relighting and also proposes training their own. Using a frozen encoder, the authors learn an adversarial decoder on multi-illumination data to perform relighting. Table 4 summaraizes the results of thier findings which is that dense pixel encoders outperform semantic encoders (eg. DINOv2) for relighting. While this intuition may in of itself not be too suprising, this paper is the first one that tests it as far as I am aware. They perform this test via their own 3 stage training process

1) First, the features from the visual encoder are added onto the predicted latent from the input image. This latent is then used to condition LuminetDiffusion to generate the reconstructed latent. This stage is regularizes via a mean regression loss in the latent space and hyperspherical regularization

2) Next, the LumiNet Diffusion model is finetuned to generate better outputs

3) Finally, supervised data is generated using the model and trained on

This method has modest improvements over prior work quantitatively and, to my eyes, much better qualitative results.

**Strengths:**

1) This paper is well written and easy to follow
3) Experiments are relatively thorough
4) Very useful insights are gained vis-a-vis lighting representations.

**Weaknesses:**

Perhaps my only reservation about this model is the use of LumiNet diffusion for generating the image. I fear the prior training of LumiNet would confound the results of the best representation. Ideally a few more decoders should be ablated.

**Questions:**

N/A see weaknesses

---

> ### Author Response · Authors · 2025-11-26
>
> **Q: Different decoder rather than LumiNet.**
>
> A: Thank you for the suggestion. We chose LumiNet as our decoder to ensure that the performance gain stems primarily from the latent representation rather than from a more powerful generative prior. Our method is compatible with other decoders, such as SD-XL, and we will include results using SD-XL in the camera-ready version.

---

### Author Response · Authors · 2025-11-26
**General response**

We thank all reviewers for their insightful feedback and thank all ACs for their great efforts during this difficult time.

**Summary**: This paper presents the first comprehensive study on visual representations for relighting, aiming to understand what makes a representation ``relightable”.  Contrary to common assumptions, our findings reveal that **stronger semantic understanding does not necessarily translate to better relighting performance.** Surprisingly, a simple representation such as **MAE outperforms more semantically powerful models like DINOv3 in relighting.** We further observe that effective relighting requires both high-level semantic and low-level photometric information. Consequently, holistic representations such as RADIOv2.5 achieve superior performance, particularly in challenging regions characterized by high ambiguity, such as metallic and glossy surfaces.

Initially, our paper received mixed scores, with two reviewers (R1 and R4) giving a score of 6 (confidence 4), and two reviewers (R2 and R3) giving scores of 2 and 4 (confidence 3 and 2, respectively).

We are pleased to see that the reviewers recognized the motivation (R2, R3), results (R3, R4), and insight (R1) of our work.

We humbly summarize the concerns of the reviewers with great respect in three parts:

1. A clear answer to the question we asked in the title: What makes a representation relightable? (raised by R2 and R4)

2. Numerical metrics and qualitative results (raised by R3 and R4)

3. Clarification of the detailed implementation.

**Response to common concerns:** We address the common concerns 1 and 2 in the general response, while concern 3 is addressed point-by-point in the individual response. We have updated a revised version of our paper to incorporate the feedback, with the changes highlighted in cyan.

**1. What makes a representation relightable? any specific aspects of the unique training strategies for RADIOv2.5 contribute to the improved performance? (R2, R4)**

Thanks for the insightful questions, we extend the discussion of what makes a representation relightable in paper (L431-476) and in the discussion section (L515-523). We echo it here again:
MAE and RADIOv2.5 are inherently better aligned with the relighting objective than contrastive features such as DINO and CLIP.
First, MAE is trained with a pixel-reconstruction objective, which tightly couples its feature space to the input image. As a result, MAE features preserve fine-grained color, shading, and reflectance information that is critical for modeling illumination. In contrast, contrastive methods like DINO and CLIP are explicitly optimized to be invariant to color, local contrast, and subtle appearance changes, largely due to their strong color and contrast jitter augmentations. These invariances suppress exactly the specular and color variations that encode material and lighting, making such features fundamentally less suitable for high-fidelity relighting. This observation is consistent with concurrent work RAE [1], which shows that an MAE-based representation with a diffusion decoder outperforms a DINO-based counterpart under the same diffusion-based decoding architecture for reconstruction — an important component for relighting.
Second, RADIOv2.5 further strengthens relighting performance. Its multi-resolution training enforces pixel-level consistency across scales, improving the reconstruction of local lighting and material effects, while its multi-model distillation introduces complementary high-level semantic and structural cues. Consequently, RADIOv2.5 features are both pixel-aligned and semantically rich—two properties that are essential for relighting, which simultaneously requires low-level appearance fidelity and high-level scene understanding.

**2. Numerical metrics and qualitative results. (R3, R4)**

As discussed in L369–374, standard metrics correlate weakly with perceived lighting quality and tend to emphasize low-frequency tone over directional effects. Please also see [2] for a detailed evaluation on how these metrics poorly correlate with human judgment. To better demonstrate the advantages of our approach, we provide qualitative comparisons in Fig. 3. In addition, we have now included a user study, which clearly shows that our method yields a significant improvement over existing approaches.

**Reference**

[1] Zheng, et al. Diffusion Transformers with Representation Autoencoders. Arxiv 2025.

[2] Giroux, et al. Towards a perceptual evaluation framework for lighting estimation. ICCV 2023.

We thank all reviewers and ACs again for their engagement and efforts.

---

### Meta-Review · Area_Chair_npyj · 2026-01-01

**Summary:**

The paper received mixed initial reviews, with scores of 6, 6, 4, and 2. Reviewers generally acknowledged that the paper is thoughtfully written and raises an interesting question about the relationship between visual representations and relighting performance. The empirical observation that highly semantic representations (e.g., CLIP- or DINO-style features) underperform for relighting, while denser, pixel-aligned representations perform better, was seen as intriguing. However, reviewers raised substantial concerns regarding the overall strength and consistency of the results, as well as the clarity and scope of the central contribution.

In the rebuttal, the authors provided clarifications and additional analysis, including an expanded discussion of the semantic–photometric trade-off, a user study to supplement standard quantitative metrics, and more detailed explanations of the training stages and feature fusion design. These responses addressed several clarity- and presentation-related concerns and helped articulate the authors’ intended perspective more clearly. That said, some key concerns remain only partially addressed. In particular, the AC anticipates that the final reviews will likely remain mixed, with one to two reviewers remaining on the negative side (including a possible retention of the score of 2) and one to two reviewers remaining mildly positive with scores of 6 (see detailed discussion in Reviewer Concerns and Reviewer Scores).

From the AC’s perspective, while the paper raises an interesting question and contains useful empirical observations, it does not yet provide a sufficiently conclusive or general answer to what makes a representation relightable. The conclusions appear highly dependent on the specific relighting pipeline and training design, and the empirical evidence does not fully support broad claims about representation properties in a pipeline-agnostic sense. Combined with shared concerns about result quality and the strength of quantitative improvements, the AC finds that the contribution does not meet the bar for acceptance in its current form. The AC therefore recommends rejection.

**Reviewer Concerns:**

### Reviewer Ru4M (Score: 6)

- The reviewer raised concerns about whether the observed relighting improvements could be confounded by the use of a strong pretrained diffusion decoder (LumiNet), rather than stemming from the representation itself. They questioned whether the conclusions about relightable representations would hold under different decoder choices.
- In the rebuttal, the authors responded briefly by stating that LumiNet was intentionally fixed across experiments to control for decoder effects, and more decoders can be included in the final version.

---

### Reviewer aB99 (Score: 2)

- The reviewer raised concerns about conceptual clarity and contribution, arguing that the paper does not clearly answer what makes a representation relightable and instead presents a relighting pipeline with feature fusion. They questioned the novelty of the feature injection approach and found the analysis of representations to be superficial. Additional concerns included confusion about quantitative results that did not consistently outperform prior work and a lack of deeper explanation beyond backbone-level comparisons.
- In the rebuttal, the authors clarified the central claim around the semantic–photometric trade-off, expanded discussion explaining why semantic invariance harms relighting, clarified the interpretation of quantitative tables, and added a user study to complement metrics that correlate weakly with perceptual relighting quality.

---

### Reviewer oftv (Score: 4)

- The reviewer raised concerns about unclear experimental protocols, ambiguity in comparisons to Latent Intrinsics, questionable benefits and potential artifacts introduced by Stage-3 training, unclear improvements over prior work, lack of cross-dataset generalization experiments, and insufficient justification of the simple feature fusion design.
- In response, the authors clarified experimental setups and evaluation protocols, corrected presentation issues, clarified and contextualized comparisons to Latent Intrinsics and prior methods, justified Stage-3 training with additional qualitative results and user studies, added fusion ablations, and expanded discussion around evaluation metrics and experimental scope.

---

### Reviewer MbbU (Score: 6)

- The reviewer requested clearer explanation of why certain representations are relightable, and inclusion of user studies, more detailed descriptions of fusion mechanisms, computational complexity and runtime comparisons, and clearer writing and presentation.
- In the rebuttal, the authors expanded the discussion of representation properties, added encoder-size ablations and runtime comparisons, conducted a user study, clarified fusion and latent extrinsics details, and improved figures and overall presentation.

**Reviewer Scores:**

### Reviewer Ru4M

- **Original score:** 6
- **Predicted final score:** 4–6
- **Rationale:** The response to this reviewer was relatively brief, and experiments with alternative decoders were promised but not fully presented or discussed, which appears to be a point of interest for this reviewer. As a result, while the reviewer may maintain their current score, there is also a possibility of a decrease.

---

### Reviewer aB99

- **Original score:** 2
- **Predicted final score:** 2–4
- **Rationale:** The rebuttal directly engages with the reviewer’s core concerns regarding conceptual clarity and interpretation, adding expanded discussion and user-study evidence. However, given the reviewer’s strong initial skepticism, it is unlikely that the rebuttal will substantially alter their overall assessment, and some reservations are likely to persist.

---

### Reviewer oftv

- **Original score:** 4
- **Predicted final score:** 4–6
- **Rationale:** Many of the reviewer’s technical and clarity-related concerns were addressed through additional explanations, ablations, and user-study evidence. However, concerns about the strength and consistency of the quantitative improvements may remain. As a result, a moderate score increase is possible but not guaranteed.

---

### Reviewer MbbU

- **Original score:** 6
- **Predicted final score:** 6
- **Rationale:** The rebuttal provides the additional details requested by the reviewer, including user studies, runtime and complexity analysis, and clearer explanations of the method. The reviewer is therefore likely to retain their already positive evaluation.

---

### Decision · Program_Chairs · 2026-01-26

Reject